# The Increase in the Drug Resistance of Acute Myeloid Leukemia THP-1 Cells in High-Density Cell Culture Is Associated with Inflammatory-like Activation and Anti-Apoptotic Bcl-2 Proteins

**DOI:** 10.3390/ijms23147881

**Published:** 2022-07-17

**Authors:** Margarita Kobyakova, Yana Lomovskaya, Anatoly Senotov, Alexey Lomovsky, Vladislav Minaychev, Irina Fadeeva, Daria Shtatnova, Kirill Krasnov, Alena Zvyagina, Irina Odinokova, Vladimir Akatov, Roman Fadeev

**Affiliations:** 1Institute of Theoretical and Experimental Biophysics, Russian Academy of Sciences, 142290 Pushchino, Moscow Region, Russia; ritaaaaa49@gmail.com (M.K.); yannalomovskaya@gmail.com (Y.L.); a.s.senotov@gmail.com (A.S.); lomovskyalex@gmail.com (A.L.); vminaychev@gmail.com (V.M.); aurin.fad@gmail.com (I.F.); shtatnovady@gmail.com (D.S.); kirill.krasnov64@gmail.com (K.K.); alennazvyagina@gmail.com (A.Z.); odinokova@rambler.ru (I.O.); vladimir.akatov@gmail.com (V.A.); 2Pushchino State Institute of Natural Science, 142290 Pushchino, Moscow Region, Russia

**Keywords:** acute myeloid leukemia, drug resistance, DNA-damaging drugs, izTRAIL, inflammation, high-density cell culture

## Abstract

It is known that cell culture density can modulate the drug resistance of acute myeloid leukemia (AML) cells. In this work, we studied the drug sensitivity of AML cells in high-density cell cultures (cell lines THP-1, HL-60, MV4-11, and U937). It was shown that the AML cells in high-density cell cultures in vitro were significantly more resistant to DNA-damaging drugs and recombinant ligand izTRAIL than those in low-density cell cultures. To elucidate the mechanism of the increased drug resistance of AML cells in high-density cell cultures, we studied the activation of Bcl-2, Hif-1alpha, and NF-kB proteins, as well as cytokine secretion, the inflammatory immunophenotype, and the transcriptome for THP-1 cells in the low-density and high-density cultures. The results indicated that the increase in the drug resistance of proliferating THP-1 cells in high-density cell cultures was associated with the accumulation of inflammatory cytokines in extracellular medium, and the formation of NF-kB-dependent inflammatory-like cell activation with the anti-apoptotic proteins Bcl-2 and Bcl-xl. The increased drug resistance of THP-1 cells in high-density cultures can be reduced by ABT-737, an inhibitor of Bcl-2 family proteins, and by inhibitors of NF-kB. The results suggest a mechanism for increasing the drug resistance of AML cells in the bone marrow and are of interest for developing a strategy to suppress this resistance.

## 1. Introduction

Acute myeloid leukemia (AML) remains a hematologic malignancy with extremely low cure and survival rates [1]. Only 29.5% of patients that are diagnosed with AML currently achieve 5-year survival, despite the overall progress in therapeutic strategies over the past decade [2,3]. To date, in the treatment of acute myeloid leukemia, in addition to chemotherapy, it is possible to use targeted therapy, immunotherapy, and the use of immune control inhibitors [1,4,5,6,7]. However, the emergence of resistance to the above treatments in leukemia cells (drug resistance), mainly localized in bone marrow niches, is the main reason for the development of relapses and, in general, the failure of AML therapy [8,9,10,11,12]. It is known that during the development of hematological malignant neoplasms, the tumor itself and surrounding areas are chronically damaged tissue with the development of chronic inflammation [12]. The development of chronic inflammation can contribute to the avoidance of tumor cell death, mediated by factors of the immune system and the action of anticancer drugs [12,13,14,15]. Critical roles of cytokines in the context of inflammation gain special interest. Pro-inflammatory mediators such as IL-1β, TNF-α, and IL-6 tend to increase in AML aggressiveness [16]. The dysregulation of the complex interactions between pro- and anti-inflammatory cytokines in AML may create a pro-tumorigenic microenvironment with effects on leukemic cell proliferation, survival, and drug-resistance [16]. The formation of drug resistance that is mediated by pro-inflammatory cytokines can be realized through an increase in the content of anti-apoptotic proteins in AML cells [17]. Bone marrow malignancy is accompanied by increased cellularity (hypercellularity), where the proportion of leukemic blasts can be greater than 60% [18,19]. In vitro studies have shown that the number of leukemic blasts can directly affect the effectiveness of chemotherapy, since the cytotoxic activity of many chemotherapy drugs depends on the number of tumor cells [20,21,22,23,24]. It is assumed that the emergence and/or increase in drug resistance in high-density cell cultures may be associated with a change in the penetration of substances into cells [25], a change in the proliferative activity of cells [26,27,28], and also with cell hypoxia [29]. However, our knowledge about the mechanism of AML cells’ increase in drug resistance in high-density cell cultures remains unclear.

The aim of this work was to elucidate the mechanism of the increase in drug resistance of acute myeloid leukemia cells in high-density cell cultures. This mechanism is of interest for suppressing the drug resistance of AML cells in bone marrow.

## 2. Results

### 2.1. Reversible Increase in the Resistance of THP-1 Cells in High-Density Cell Cultures to DNA-Damaging Drugs and izTRAIL

We found an increase in the resistance of THP-1 cells in high-density cultures after 5 days of cultivation to both DNA-damaging drugs and the recombinant antitumor cytokine izTRAIL. Micrographs of THP-1 cell cultures at 24 h (low-density cultures, LDC) and 120 h (high-density cultures, HDC) after cell seeding are shown in Appendix A.

The IC50 values (50% inhibitory concentration) were 1.2 ± 0.5 μM for etoposide; 0.05 ± 0.01 μM for topotecan; 1.9 ± 0.5 μg/mL for cytarabine; 5.3 ± 0.3 µM for cisplatin; 0.22 ± 0.01 µM for doxorubicin; and 0.019 ± 0.005 µg/mL for izTRAIL for THP-1 cultures at 1 day after cell seeding (hereafter referred to as THP-1LDC, 10^5^ cells/mL). In turn, for THP-1 cells after 5 days of cultivation (hereafter THP-1HDC, 10^6^ cells/mL), the IC50 values were 100 ± 5 μM for cisplatin, and 16.2 ± 1.1 μM for doxorubicin, and at high concentrations of etoposide (50 μM); topotecan (16 µM, cytarabine (26 μg/mL); and izTRAIL (1.5 μg/mL), the number of live cells was not less than 70%, relative to the control (Figure 1). A similar increase in drug resistance for other AML cells such as HL-60, MV4-11, and U937 was also revealed (Appendix A).

The increase in the drug resistance of cells in dense cultures is reversible. Figure 2 shows that cells that are taken from THP-1HDC cultures and seeded at a low density lose the increased resistance characteristic of THP-1HDC cells during a day.

Thus, the results that are presented show the reversable increase in drug resistance of AML cells in high-density cell cultures.

### 2.2. The Increase in Drug Resistance of THP-1 Cells in High-Density Cultures May Depend Not Only on Culture Density

To assess the factors that determined the increase in AML cells’ drug resistance in high-density cultures, we compared the cytotoxic effect of various drugs on THP-1HDC cells and on THP-1 cells 1 day after their inoculation, at a concentration at which the culture density by this time would be the same (1 × 10^5^ cells/well) as in THP-1HDC (hereafter referred to as THP-1HDC_1day_). The increase in the resistance of THP-1 cells to cytarabine and cisplatin in THP-1HDC_1day_ cultures, compared to the control, was the same as in THP-1HDC (Figure 3). However, THP-1HDC_1day_ cells were significantly more sensitive to doxorubicin (IC50 = 2.8 ± 0.4 µM), izTRAIL (IC50 = 0.15 ± 0.01 µg/mL), and especially to etoposide (IC50 = 1.2 ± 0.5 µM) and topotecan (IC50 = 0.26± 0.01 µM) than THP-1HDC cells (Figure 3).

The results suggest that the increase in the drug resistance of AML cells in high-density cell cultures is determined not only by the density of the cell culture, but also by the residence time of cells in such cultures.

### 2.3. The Increase in Drug Resistance of THP-1HDC Cells Occurs While Maintaining Their Proliferative Activity and Is Not Associated with Cell Hypoxia

The mechanism of the increase in the drug resistance of tumor cells in high-density cell cultures is of interest in many studies [20,21,22,23,24]. One of the main reasons for this increase in drug resistance is supposed to be the inhibition of cell proliferation in dense cell cultures [26,27,28]. However, in our study, proliferation of the resistant THP-1HDC cells continued and the percent of dead cells was low (Appendix A). The proliferation of THP-1HDC cells was also confirmed by cell cycle analyses (Figure 4a), although the percentage of THP-1HDC cells increased in G1/G0 from 49 ± 3% to 59 ± 3% (*p* < 0.05) and decreased in phase S from 43 ± 3% to 33 ± 2% (*p* < 0.05), compared to THP-1LDC cells. The percent of Ki-67 positive cells and the percent of mitotic cells in THP-1HDC (98 ± 2% and 1.4 ± 0.2%) and THP-1LDC (99 ± 2% and 1.8 ± 0.3%) cultures did not differ significantly, pointing to the proliferation of THP-1HDC cells as well.

Hypoxia can induce the drug resistance of tumor cells in high-density cell cultures via anti-apoptotic proteins [30]. Therefore, we evaluated the hypoxia in resistant THP-1HDC cells. Cell hypoxia was assessed by the level of HIF-1α protein expression, as well as cell staining with the hypoxia green reagent probe. THP-1 cells, which were grown at a reduced content of O_2_ (1%) in the gas mixture for 1 day (hereafter THP-1LDC1%O_2_) were used as a positive control.

Figure 4b,c show that in THP-1HDC cells, both the level of expression of the HIF-1α protein and the mean intensity of the fluorescence of hypoxia green reagent did not differ significantly (*p* ≥ 0.05) from these values of THP-1LDC cells. In turn, when THP-1 cells were grown at 1% O_2_ in a gas mixture at 37 °C for 1 day, the expression level of HIF-1α protein cells increased more than three times, and the average fluorescence intensity of hypoxia green reagent increased more than 10 times, compared to THP-1LDC cells. Thus, hypoxia did not occur in THP-1HDC cells, which appeared to increase drug resistance.

Thus, the increase in the drug resistance of THP-1 cells in high-density cell cultures occurred while maintaining cell proliferation and without the signs of cell hypoxia.

### 2.4. Comparative Bioinformatics Analysis of the Transcriptomes of THP-1HDC and THP-1LDC Cells

To study the mechanism of the increase in drug resistance in THP-1 cells in high-density cell cultures, the comparative analyses of the transcriptomes in THP-1HDC and THP-1LDC cells were performed.

First, we explore the data of RNA seq for finding patterns of difference and similarities in the two groups using a principal component analysis (PCA). The PCA plot for the first and second principal components is shown in Appendix A. There is a clear difference between THP-1HDC and THP-1LDC cells in the first principal component, which explains 57% of the variance. 

Differentially expressed genes (DEG) were initially screened based on the requirement that a fold change (FC) of genes in the compared cells must correspond to the range Log_2_FC ≥ 1, Log_2_FC ≤ −1, *p* < 0.05. Then we performed a functional analysis and an analysis of pathway enrichment with differentially expressed genes.

Figure 5 shows the 10 main GO (gene ontology) terms of categories BP (biology processes); MF (molecular functions); and CC (cellular components) for DEG. The up-expressed genes were mainly associated with the upregulation of I-kappaB phosphorylation (GO:1903721) and the regulation of the chronic inflammatory response (GO:0002676) in the BP category; stearoyl-CoA-9-desaturase activity (GO:0004768) and acyl-CoA desaturases (GO:0016215) in the MF category; and constituted mainly specific granular membrane (GO:0035579) and tertiary granule (GO: 0070820) in the CC category. An analysis with the KEGG (Kyoto Encyclopedia of Genes and Genomes) pathway showed that the up-expressed genes were mainly enriched in the unsaturated fatty acid biosynthesis signaling pathway (KEGG:01040). Biological processes such as the up-regulation of I-kappaB phosphorylation and regulation of the chronic inflammatory response are among the key players in the inflammation process [31]. Stearoyl-CoA-9 and acyl-CoA desaturases are involved in the inflammatory process. In particular, they are involved in the synthesis of polyunsaturated acids, such as arachidonic acid, and are precursors of inflammatory mediators, such as eicosanoids [32,33,34].

Furthermore, we found that downregulated genes were mainly involved in the regulation of zinc ions transmembrane transport (GO: 0071580) in the BP category; in the binding of leptomycin B (GO:1901707) and to the galanin receptor (GO:0031763) in the MF and Golgi category; and to plasma membrane transport vesicle (GO:0070319) and amino acid transport complex (GO:1990184) in category CC. An analysis of the KEGG pathway showed that downregulated genes mainly enriched pathways that were associated with mineral uptake (KEGG:04978) (Figure 5b).

The regulation of transmembrane transport, together with such molecular functions as leptomycin B binding or binding to the galanin receptor, and the mineral uptake signaling pathway (mediated primarily by metallothioneins), are associated with processes that aim to suppress the inflammatory response [35]. For example, leptomycin B has been shown to act as an inhibitor of the stabilization of cyclooxygenase 2 mRNA, induced by IL-1β [36]; the cytoplasmic expression of IFN-alpha1 mRNA [37]; and NF-kB activity, due to an increased accumulation of the NF-κB inhibitor in the nucleus [38]. In turn, galanin can regulate the secretory profile of macrophage cytokines and chemokines, depending on their differentiation and polarization [39]. In vitro studies have shown that zinc can reduce the activation of NF-κB and its target genes, such as TNF-α and IL-1β, and increase the expression of the A20 and PPAR-α genes, two zinc finger proteins with anti-inflammatory properties [40]. In turn, metallothioneins, which play an important role in the absorption, distribution, storage, and release of metals, can capture a wide range of reactive oxygen species (ROS) that are induced by TNF-α [40].

Thus, the results of the functional analysis and analysis of the enrichment of pathways with differentially expressed genes indicate the activation of the genes that are involved in the inflammatory response and a decrease in the activity of the genes that are suppressed in the inflammatory response in high-resistant THP-1HDC cells, compared to THP-1LDC cells. 

A network analysis of protein–protein interactions (PPI) for differentially expressed genes was performed in Online STRING. We found 92 nodes with 85 connections in the up-regulated PPI gene network and 29 nodes with 16 connections in the down-regulated PPI gene network (Figure 6a,b). In addition, we examined hub genes based on their higher connectedness values in the PPI network. It was found that in THP-1HDC cells, TNFα is the hub node in the network of interaction between the proteins of up-regulated genes, which has more than 10 links. The product of this gene is the pleiotropic cytokine TNF-a, the main function of which is the regulation of all stages of the inflammation process [41]. We did not find a hub node of the network of interaction between the proteins of down-regulated genes.

To study the main functions of the genes that are hub nodes of the PPI network, we performed a functional analysis of annotations based on the GO terms using the ClueGO plugin for Cytoscape. Statistical parameters for the ClueGO enrichment analysis were established based on a two-tailed hypergeometric test with *p* ≤ 0.05 and a Kappa score of ≥0.4 as the primary criterion, and advanced parameters of GO three interval levels 3–8, GO term/pathway selection = min. 1 gene and percent of genes = 4%. The biological processes’ gene ontology of the TNF hub gene of the PPI network was predominantly enriched in the positive regulation of the chronic inflammatory response (Figure 6c).

The results indicate that TNFa is the major differentially expressed gene that activates the inflammation-like genotype and positive regulation of the chronic inflammatory response in THP-1HDC cells.

It is known that the TNF-induced nuclear factor NF-κB plays a critical role in the formation of the inflammatory response [42,43]. In this regard, the gene set enrichment analyses (GSEA) was made for drug resistant THP-1HDC cells, compared to the drug-sensitive THP-1LDC cells in NF-kB-regulated signaling pathways that are responsible for the inflammatory reaction. We used the sets of genes that comprise the main traits (the Hallmarks collection). The analysis showed that in THP-1HDC cells, compared to THP-1LDC, the set of NF-kB regulation genes in response to TNF and the set of inflammatory response genes from the HALLMARK collection were enriched (Figure 7).

In general, the obtained data indicated that the increase in the drug resistance of THP-1 cells in high-density cultures occurred upon activation of the signaling pathways that were associated with the inflammatory response.

### 2.5. THP-1HDC Cells Demonstrate Signs of Inflammatory-like Activation

The findings of the bioinformatic analysis of transcriptome suggest that the activation of the genes that are associated with the inflammatory response in THP-1HDC cells is the cause of their increased drug resistance. This possibility is supported by our data on a significant increase in drug resistance in THP-1LDC cells that are treated with the pro-inflammatory agent LPS (THP-1LPS cells) (Appendix A).

To assess the inflammatory-like activation of THP-1HDC cells, the expression of CD markers that were associated with inflammation, the production of intracellular nitric oxide (NO), as well as the secretion of cytokines were analyzed. THP-1LPS cells were used as a positive control for the inflammatory-like phenotype. The obtained data show a significant increase in the percentage of THP-1HDC cells bearing CD markers on their surface, indicating the formation of an inflammatory-like phenotype, compared to THP-1LDC cells (Figure 8a). The percentage of CD68 and HLA-DR positive cells increased significantly (*p* ≤ 0.01) in the THP-1HDC cell population, relative to THP-1LDC cells. The cells carrying integrin αM (CD11b) and integrin αX (CD11c) (24 ± 1% and 35 ± 1%, respectively) were shown to appear in the population of THP-1HDC cells. Similar immunophenotype changes in THP-1LPS cells as in THP-1HDC were found (Figure 8a). However, THP-1HDC cells were negative for the markers CD14, CD36, CD54, and CD284 in contrast to THP-1-LPS cells (Appendix A).

A change in intracellular nitric oxide production is a sign of the acquisition of a pro-inflammatory-like phenotype by monocytic cells [44]. It was found that the level of intracellular NO decreased significantly in THP-1HDC and THP-1LPS cells (*p* < 0.01 and *p* < 0.05, respectively), compared to THP-1LDC cells (Figure 8b).

Therefore, a decrease in NO production was revealed in THP-1HDC cells, which is typical for cells with an inflammatory-like activation.

The specificity of cytokine secretion by cells may reflect their inflammatory activation. We studied the secretion of cytokines by THP-1LDC, THP-1HDC, and THP-1LPS cells. THP-1LPS were chosen as control cells that were activated by a pro-inflammatory agent (LPS). Figure 8c shows the accumulation of some cytokines in growth medium in the cultures that are noted above. The numerical values of the concentrations of analyzed cytokines in cultures THP-1LDC, THP-1HDC, and THP-1LPS are presented in Appendix A. THP-1HDC and THP-1LPS cells were characterized by the de novo secretion of IL-4, eotaxin, basic FGF, G-CSF, MIP-1α, IP-10, and by increased secretion of IL-1β, IL-1ra, IL-6, Rantes, TNF-α, IL-9, MCP-1, and MIP-1β, compared to THP-1LDC. At the same time, the constitutive secretion of IL-2, IL-5, IL-8, IL-12p70, IL-15, IL-17A, IFN-γ, PDGF-BB, and VEGF by THP-1HDC cells was decreased, relative to THP-1LDH cells. The secretion of pro-inflammatory cytokines in the THP-1HDC culture was increased for some of them (IL-beta, IL-6, Rantes, TNF-α, IP-10, MCP-1, MIP-1α, MIP-1β, bFGF, G-CSF, PDGF BB) and decreased for others (IL-2, IL-5, IL-7, IL-8, IL-12p70, IL-17A, IFN-gamma) as compared to THP-1LDC. In THP-1HDC culture, both an increase (IL-1ra, IL-4, IL-9) and a decrease (IL-10, IL-13) in the secretion of anti-inflammatory cytokines were also revealed (Appendix A). The secretion of cytokines in THP-LPS culture significantly exceeded those in THP-1LDC and THP-1HDC, except Rantes, which was mostly actively secreted in the THP-1HDC culture.

The results indicate that THP-1 cells in high-density cultures acquire phenotypic characteristics with signs of inflammatory-like activation. In addition, the results suggest that the increase in the drug resistance of THP-1 cells in high-density cultures is caused by the activation of signaling pathways related to an inflammatory response and cell survival.

### 2.6. The Transcription Factor NF-kB Is Activated and the Content of Anti-Apoptotic Proteins Bcl-2 and Bcl-xL Increases in THP-1HDC Cells 

Transcription factor NF-kB is known to be the marker of the inflammatory activation of cells [45,46]. Therefore, we studied the activity of the transcription factor NF-kB and analyzed in cells the content of the anti-apoptotic proteins Mcl-1, Bcl-2, Bcl-xL, Bcl-w, and Bfl-1 which are under the control of NF-kB [47]. 

The transcription factor NF-kB is upregulated in THP-1HDC, compared to THP-1LPS cells. The content of phosphorylated RelA (p65), the main marker of NF-kB transcriptional activity, increased significantly in high-resistant THP-1HDC and THP-1LPS cells relative to control THP-1LDC cells (Figure 9a). The content of anti-apoptotic proteins Bcl-2 and Bcl-xL, but not Mcl-1, Bcl-w, and Bfl-1, was significantly increased in THP-1HDC cells, compared to THP-1LDC cells. However, another set of anti-apoptotic proteins, Mcl-1 and Bcl-xL, were overexpressed in THP-1LPS cells, but not Bcl-2, Bcl-w and Bfl-1 (Figure 9).

Therefore, an increase in the resistance of THP-1 cells in high-density cell cultures occurs together with the activation of the transcription factor NF-kB and with an increase in the content of the anti-apoptotic proteins Bcl-2 and Bcl-xL.

### 2.7. The Inhibitors of NF-kB and Anti-Apoptotic Proteins of the Bcl-2 Family Contribute to Suppression of the Increased Drug Resistance in THP-1HDC Cells

Specific inhibitors of the transcription factor NF-kB (NF-kB Activation Inhibitor IV, JSH-23 and QNZ) and of the anti-apoptotic proteins of the Bcl-2 family (ABT-737) were used to suppress the increased drug resistance of THP-1HDC cells. The 4-h preincubation of cells with the inhibitors was provided before drug addition. Figure 10 shows that inhibitors IV and JSH-23, more pronounced than QNZ, suppressed both the activation of NF-kB and the resistance of THP-1HDC cells to etoposide and izTRAIL. These inhibitors also suppressed the increased resistance of THP-1HDC cells to topotecan and, to a lesser extent, to cytarabine (Appendix A). ABT-737 showed the strongest inhibition of resistance of THP-1HDC cells to doxorubicin, izTRAIL, topotecan, and etoposide, and to a lesser extent to cytarabine. None of the inhibitors that were used suppressed the resistance of THP-1HDC cells to cisplatin (Appendix A).

The results indicate the participation of NF-kB and anti-apoptotic proteins of the Bcl-2 family in the mechanism of the increasing drug resistance of THP-1 cells in high-density cell cultures and a large variability of this participation for different drugs.

## 3. Discussion

The role of the microenvironment in the drug resistance of AML cells is of scientific and applied importance and requires an understanding of its mechanisms [12,48,49]. The increase in drug resistance in leukemic cells in high-density cultures was marked earlier [50,51]; however, the mechanism of this phenomena remains unclear. The results of our work show the great increase in the resistance of different AML cells (lines HL-60, THP-1, MV411, U937) in high-density cultures to inhibitors of topoisomerases, etoposide and topotecan [52,53]; an antimetabolite, cytarabine [54]; DNA targeting antitumor drugs doxorubicin and cisplatin [55,56,57]; as well as to antitumor recombinant protein izTRAIL [58]. It was important that the increase in the drug resistance of AML cells in high-density cell culture was reversible and that the transfer of the AML cells from high-density to low-density cultures decreased the resistance to an initial state. These results indicate the transient epigenetic mechanisms of the increased drug resistance of AML cells in high-density cell cultures. Another interesting result is that the increase in the drug resistance of AML cells depends not only on the cell culture density, but also on the residence time of cells in these cultures. At the same time, the increase in the drug resistance of AML cells can form quickly for some drugs (cisplatin, cytarabine), and slowly for others (etoposide, topotecan, doxorubicin, and izTRAIL). These data point to the prolonged activation of different pathways of cell survival during the change of cell microenvironment in culture. 

The results of our study show that the mechanisms of the increase in the resistance of AML cells in high-density cultures to various drugs can differ. The sign of this difference is that the maximum resistance was revealed 24 h after seeding the high-density cultures of THP-1 cells to cisplatin and cytarabine, but not to etoposide, topotecan, doxorubicin, and izTRAIL. The largest resistance of THP-1 cells in high-density cultures to etoposide, topotecan, doxorubicin, and izTRAIL appeared only after a long-term cultivation. These results also indicate that, in general, the increase in the drug resistance of THP-1 cells in high-density cultures is not associated with a smaller amount of drug per cell, compared to low-density cultures.

According to some studies, the suppression of proliferative activity may be the reason for the increase in the drug resistance of AML cells in high-density cultures [26,27,28,59]; however, our data do not suggest this mechanism for increasing drug resistance in THP-1 cells in high-density cell cultures. The cells in the high-density cell cultures THP-1HDC grew, and indicators of their proliferation (growth rate of cell number, Ki67, mitotic index, and the percent of cells in G1 and S phases of cell cycle) were similar to those in the low-density THP-1LDC cultures.

Hypoxia may be suggested as another reason for increased drug resistance in THP-1HDC cells [30]. However, we revealed no hypoxia in these cells.

The results of our bioinformatics analyses of transcriptome point to the activation of signaling pathways that are associated with the formation of a chronic inflammatory response and the activation of I-kappa B phosphorylation in drug-resistant THP-1HDC cells, compared to THP-1LDC. The analysis showed that the set of NF-kB regulation genes in response to TNF and the set of inflammatory response genes from the HALLMARK collection were enriched in THP-1HDC cells, in contrast to THP-1LDC. In addition, we found that TNF (which regulates inflammation processes, as well as the activation of the genes of signaling pathways that are regulated by NF-kB in response to TNF, and is also responsible for the inflammatory activation) is the major differentially expressed gene in THP-1HDC cells. Chronic inflammation is known to develop in the tumor microenvironment, which can help malignantly transformed cells to avoid the death that is mediated by factors of the immune system and the action of anticancer drugs [12,14,15]. It is known also that the increase and/or emergence of drug resistance in tumor cells may be accompanied by the activation of signaling pathways that are associated with the inflammatory response [42,60,61,62,63,64], which is consistent with the results of our work. Our data suggest a high cell culture density of tumor cells as a cause of the pro-inflammatory change in their microenvironment and increase in their drug resistance.

The inflammatory-like activation of THP-1HDC cells was confirmed by the study of NO production in cells and by an immunophenotype analysis, provided in comparison with THP-1 cells that were treated with inflammatory agent LPS (THP-1LPS). However, the similarity and the difference in the change of CD markers on THP-1HDC and THP-1LPS cells should be noted. Some markers appeared de novo (CD11b, CD11c) or overexpressed (CD 68, HLADR) on both cell types, while others appeared only on THP-1LPS cells (CD14, CD36, CD54, CD284). The reason for the difference may be that LPS is a factor in acute inflammation, and a chronic inflammatory response is formed in THP-1HDC cells, according to the bioinformatics analyses of transcriptome. A similar situation was revealed in the change of cytokine secretion by THP-1HDCand THP-1LPS cells. The secretion of all the 27 cytokines that were analyzed was increased in THP-1LPS cells, compared to THP-1LDC, while in THP-1HDC cultures, the secretion of 10 cytokines was reduced and the secretion of 17 cytokines was increased, including those that were important for inflammatory activation, such as TNF-α, IL-1β, IL-6, Rantes, IL-9, MCP-1, MIP-1β, and IP-10 [16,65,66,67,68,69,70]. Chronic inflammation in the tumor microenvironment is known to be supported by the synthesis of inflammatory mediators (cytokines and chemokines) by both stromal and immune cells and leukemic blasts [71,72]. The change in the set of cytokines in AML can create not only a pro-oncogenic microenvironment, but also the modulate proliferation, survival, and drug resistance of leukemic cells [16]. Our data show that an inflammatory environment can be formed by AML cells themselves, because of the high density of cell cultures.

Our results show that the inflammatory-like activation that is induced by a change in the secretion of cytokines in high-density cell cultures of THP-1HDC cells activates the NF-kB-related signaling pathways and anti-apoptotic proteins of the Bcl-2 family, including Bcl-2 and BCL-xl, but not Mcl-1, Bfl-1, and Bcl-w. It is interesting that the inflammatory activation of THP-1 cells by LPS induced an increase in the content of Mcl-1 and Bcl-xl only. This fact also indicates different mechanisms of the inflammation-like activation of AML THP-1 cells in high-density cell cultures, and by LPS. Noteworthy, an increase in the drug resistance of the cells in high-density cell cultures was significantly more potent than in THP-1LPS cells for all the drugs that were tested, except izTRAIL. The increase in the resistance to izTRAIL in both THP-1HDC and THP-1LPS cells was the same and great.

The low effects of the inhibitors of NF-kB on the increased drug resistance of THP-1HDC cells together with their effective suppression of IkB phosphorylation in the cells indicate that this transcription factor is involved in the mechanism of the drug resistance of THP-1 cells in high-density cell cultures, but only weakly. Therefore, NF-kB-related signaling pathways do not determine the major mechanisms of the increase in drug resistance in high-density cell culture of THP-1 cells. The most potent suppression of drug resistance in THP-1HDC cells was revealed with ABT-737, indicating the significant association of the increase in the resistance of THP-1 cells in high-density cell cultures to different drugs with anti-apoptotic proteins of the Bcl-2 family. No effects of ABT 737 and NF-kB inhibitors on the increased resistance of THP-1HDC cells to cisplatin point to activation of the mechanism of the resistance to this drug, which is independent of both the factor NF-kB and Bcl-2 proteins. Taken together, these data indicate the need to elucidate the mechanisms of increased drug resistance of AML cells in high-density cell culture to each drug, in order to suppress this resistance. We can suggest that the studied phenomenon reflects the situation in bone marrow during the active growth of blasts. Our results suggest that the use of inhibitors of the Bcl-2 family of anti-apoptotic proteins may be an effective strategy to sensitize AML cells to drug treatment in patients with extremely high white blood cell counts in the bone marrow. To summarize our results, the great increase in the resistance of AML cells in high-density cell cultures to different drugs was revealed. The obtained data suggest that this increase in drug resistance is associated with the activation of cytokine secretion by AML cells in high-density cell cultures, followed by their inflammatory activation, and results in the increase in their drug resistance via anti-apoptotic proteins of the Bcl-2 family (Figure 11). The activation of NF kB can be partly involved in this mechanism of the increase in drug resistance.

## 4. Materials and Methods

### 4.1. Chemicals

Hypoxia green reagent; pHrodo Green E. coli; DAF-FM DA; JSH-23; QNZ; ABT-263; and ABT-737 were purchased from Thermo Fisher Scientific (Waltham, MA, USA). The fetal bovine serum was from Gibco (Gibco, Sigma Aldrich Company Ltd., Waltham, MA, USA). Intracellular staining permeabilization wash buffer; Human TruStain FcX (Fc Receptor Blocking Solution); PE anti-human Ki-67; isotype control antibodies PE Mouse IgG1 k isotype Ctrl; APC Mouse IgG1 k isotype Ctrl; FITC Mouse IgG1 k isotype Ctrl; PE Mouse IgG2a k isotype; PE Mouse IgG2b k isotype and antibodies APC anti-human CD11b; PE anti-human CD284; PE anti-human CD36; PE anti-human CD33; PE anti-human HLA-DR; PE anti-human HIF-1α were from BioLegend (San Diego, CA, USA). MycoFluor™ Mycoplasma Detection Kit was from Molecular Probes Inc. (Eugene, Oregon, USA). FITC anti-human CD68 was from BD Bioscience (Franklin Lakes, NJ, USA). Culture medium RPMI 1640; 2-mercaptoethanol; PBS; Calcein AM; Bisbenzimide Hoechst 33342 (H33342); propidium iodide (PI), resazurin; LPS of *E. coli* O111: B4; antibodies FITC anti-human CD11c; FITC anti-human CD14; FITC anti-human CD54; NF-kB activation inhibitor IV, and other chemicals were purchased from Sigma-Aldrich (St. Louis, MO, USA). 

### 4.2. Protocol of izTRAIL Preparation

To obtain a soluble trimeric form of the izTRAIL protein, we synthesized the isoleucine zipper motif and the izTRAIL gene and cloned them into the plasmid vector (No-vagen, Madison, WI, USA). The resulting gene was used to transform *E. coli* BL21 (DE3) cells, and a trimeric form of izTRAIL with a molecular weight of 80 kDa was obtained by microbial synthesis, followed by purification by metal affinity chromatography [73].

### 4.3. Cell Cultures

Human acute leukemia cells (lines THP-1, MV4-11, HL-60, U-937) were obtained from the ATCC (Manassas, VA, USA). The cells were cultured in RPMI 1640/F12 medium supplemented with 10% fetal bovine serum (FBS) and 40 μg/mL of gentamicin sulfate at 37 °C in a humidified atmosphere of 5% CO_2_. THP-1LPS cells were obtained by incubation of THP-1 cells in RPMI 1640/F12 supplemented with 10% FBS and 10 µg/mL of LPS of E. Coli O111:B4 for 24 h after the cell seeding. The cell cultures were tested for mycoplasma infection using the MycoFluor ™ Mycoplasma Detection Kit, and no mycoplasma was detected.

### 4.4. Cytotoxicity Assay and Drug Treatment

The cells were taken from cultures after growth for 3 days and seeded in U-shaped wells of a 96-well plate in the amount of 5 × 10^3^ cells in 0,1 mL of growth medium per well. The recombinant protein izTRAIL or DNA damaging drugs were added to cultures 24 h (LDC) or 120 h (HDC) after cell seeding. The cytotoxicity was assessed by the ratio of the number of living cells in the experimental and control (nontreated) cultures at 24 h after drug addition. The number of living cells was evaluated by resazurin reduction [74]. The cells were incubated with resazurin (30 μg/mL) for 4 h at 37 ° C and 5% CO_2_ and then the fluorescence intensity of incubation medium was measured at Ex.532 nm/Em.590 nm using an infinite F200 plate reader (Tecan, Männedorf, Switzerland).

### 4.5. Cell Immunophenotyping

To study the expression of CD markers, the cells were harvested from culture flasks and washed in cell-staining buffer, 300 g, for 5 min. Staining was performed using a panel of monoclonal antibodies: APC anti-human CD11b; FITC anti-human CD11c; FITC anti-human CD14; FITC anti-human CD68; PE anti-human CD284; PE anti-human CD36; PE anti-human HLA-DR; and FITC anti-human CD54. To determine nonspecific binding, the cells were stained with isotype control antibodies: APC Mouse IgG1 k isotype Ctrl; FITC Mouse IgG1 k isotype Ctrl; PE Mouse IgG1 k isotype; and PE Mouse IgG2a k isotype. The staining was carried out at room temperature in the dark for 30 min. After staining, the cells were fixed with 2% paraformaldehyde solution and an analysis of CD expression was performed using a BD Accuri C6 flow cytometer (BD Bioscience, Franklin Lakes, NJ, USA) [75].

### 4.6. Cell Proliferation and Cell Viability Assays

The cells were seeded in U-shaped wells of a 96-well plate at a concentration of 5 × 10^3^ cells per well in 100 μL of growth medium and cultured in a CO_2_ incubator. The number of cells in suspension and their viability after their detachment with Accutase were analyzed using a BD Accuri C6 flow cytometer. Cell viability was assessed after staining them in suspension in culture medium with 200 nM of Calcein AM fluorescent dyes and 1 μg /mL of propidium iodide [76]. Cell proliferation was also assessed by DNA cytograms in cell populations and the expression of the Ki-67 nuclear antigen using the flow cytometer, and by the percent of mitotic cells [75]. To analyze DNA histograms, the cells were suspended in phosphate-buffered saline, fixed with 70% ethanol, and stained with 1 μg/mL of propidium iodide. To analyze the expression of Ki-67, we used PE anti-human Ki-67 antibodies. The control cells were stained with PE Mouse IgG1 k isotype Ctrl. The cell cycle was analyzed using ModFit LT 4.1 software (Topsham, ME, USA). Mitotic cells were evaluated by the fluorescence of cells that were stained with nuclear dyes H 33342 at a concentration of 1 μg/mL and counting the number of mitotic cells using a DM 6000 fluorescence microscope (Leica, Wetzlar, Germany) [75]. The total number of cells that were analyzed in randomly selected fields was at least 500.

### 4.7. Cell Hypoxia Assessment

The hypoxia in the cells was assessed by the expression of the HIF-1α protein using the PE anti-human HIF-1α and PE mouse IgG2b k isotype (BioLegend, San Diego, CA, USA), as well as by staining the cells with the Image-iT green hypoxia reagent probe (Thermo Fisher Scientific, Waltham, MA, USA) using a BD Accuri C6 flow cytometer.

### 4.8. Analysis of Genome-Wide Transcriptional Activity 

The whole-genome transcriptional activity of genes was studied by the nanopore sequencing of cDNA products that were obtained from mRNA. For RNA sequencing, suspensions of THP-1 cells of LDC and HDC (two groups) were taken, centrifuged (2 min, 4000× *g*), and 10^7^ cells of a supernatant of each group were lysed in 1 mL of the reagent ExtractRNA (Evrogen, Russia) for 15 min. The obtained cell lysates were centrifuged (10 min, 15,000× *g*) and used for RNA seq analysis in three independent experiments. Synthesis of cDNA from mRNA and amplification of the synthesized cDNA were performed using Mint and Encyclo Plus PCR kits (Evrogen, Russia), respectively. The resulting cDNA library was multiplexed using the Native Barcoding Expansion 1-12 (PCR-free) kit (Oxford Nanopore Technologies, UK). The library for sequencing was assembled based on the obtained cDNA amplicons using the Ligation Sequencing Kit (Oxford Nanopore Technologies, UK). Amplification of cDNA libraries was carried out at 20 °C—5 min and 65 °C—5 min, and the reaction mixture was purified using the AMPure XP kit (Beckman, Brea, CA, USA). To measure the amount of double-stranded DNA in the reaction mixture, a QuDye HS kit (Lumiprob, Russia) was used. Library DNA sequencing was performed using a MinION sequencer (Oxford Nanopore, UK) with a FLO-MIN106D cell (Oxford Nanopore, UK), using the MinKNOW basic software (Oxford Nanopore, UK). Basecalling and demultiplexing of the obtained data were performed using the Guppy v5.0.16 program (Oxford Nanopore, UK). Pipeline-transcriptome-de (https://github.com/nanoporetech/pipeline-transcriptome-de, accessed on 29 May 2022) was used to process the primary bioinformatic transcriptome sequencing data.

### 4.9. Bioinformatics Analyses of Transcriptome

A principal component analysis (PCA) was performed using the iDEP.951 online data-processing service for two groups, LDC and HDC, in principal component 1 (PC1) and principal component 2 (PC2) coordinates [77]. To analyze the differential expression of genes (DEG), the data table was sorted. Data with log_2_FC (Fold Change) ≥ 1 and log_2_ FC ≤ −1 were selected as DEG and opted for network construction. GO (Gene Ontology) enrichment and a KEGG (Kyoto Encyclopedia of Genes and Genomes) pathway analysis of selected DEGs were performed using ClueGO (v. 2.5.8) and CluePedia (v. 1.5.8). The analysis of gene ontology (GO) included three categories: molecular function (MF), biological processes (BP), and cellular component CC). For the statistical analysis, the t-test (ANOVA) was performed as the default setting. In the pathway functional analysis, genes were mapped to KEGG pathways, and a *p*-value of <0.05 and count of >2 were set as the thresholds [33]. The statistical significance was determined using the Bonferroni step-down test and a two-tailed hypergeometric enrichment/depletion test. The protein–protein interaction (PPI) network was constructed using the STRING (Search Tool for the Retrieval of Interacting Genes; http://string-db.org, accessed on 29 May 2022) (version 11.5) online database. The PPI network of upregulated and downregulated DEGs constructed using STRING with a confidence score of >0.4 was considered as statistically significant. Data were then imported into Cytoscape (v. 3.9.1) to visualize the protein interaction network. For identifying subgroups of genes sharing similar expression patterns across multiple conditions, a module analysis was carried out. Hub genes were determined by the number of relationships >10 using the built-in filter in Cytoscape. ClueGO is a Cytoscape plugin that can divide large clusters of genes into functional groups based on GO and KEGG. CluePedia is a ClueGO plugin that can integrate information regarding genes, proteins, and microRNAs into a network with ClueGO terms/pathways [78]. Gene set enrichment analysis (GSEA) uses the rank of the degree of differential gene expression in two types of samples to test whether the preset gene set is enriched at the top or bottom of the ranking table and interpret biological information from another angle. The Molecular Signatures Database (MSigDB) was used to perform GSEA (GSEA v4.2.2, http://www.broadinstitute.org/gsea/, accessed on 29 May 2022) [79].

### 4.10. Intracellular NO Assay

To assess the intracellular NO production, the cells were stained by incubating them with 5 μM of DAF-FM DA for 40 min, washed with a fresh growth medium and then incubated for 30 min. To study the inducible NO production, the cells were preincubated with 10 μg/mL of LPS from *E. coli* O111: B4 for 24 h. All the incubation procedures were performed in a CO_2_-incubator. Cell fluorescence was analyzed using a BD Accuri C6 flow cytometer [75].

### 4.11. Multiplex Analysis of Cytokine Production

Culture medium for the analyses was taken and centrifuged (300 *g*, 5 min) one day after cell seeding at a concentration of 5 × 10^4^ cells/mL with or without 10 µ/mL of LPS, or 5 days after the cell seeding in U-shaped wells of a 96-well plate. The cytokine content in a supernatant was evaluated with a commercial Bio-Plex Pro Human Cytokine Grp I Panel 27-plex kit [41] (Bio-Rad, Hercules, CA, USA) using a Bio-Plex MAGPIX multiplex analyzer (Bio-Rad, Hercules, CA, USA).

### 4.12. Analysis of NF-kB Activation

The activation of the transcription factor NF-kB was determined by the level of phosphorylated RelA (p65) in cells, which was measured by ELISA using a commercial kit NFkB p65 (Total/Phospho) InstantOne TM ELISA (Thermo Fisher Scientific, Waltham, MA, USA).

### 4.13. Western Blotting Analysis

Whole cell lysates were prepared using RIPA reagent (Santa Cruz, Dallas, TX, USA). A total of 30 µg of protein from each sample was resolved by SDS-polyacrylamide gel electrophoresis (PAGE) and transferred to a 0.2 μm nitrocellulose membrane (Bio-Rad, Hercules, CA, USA). The membranes were blocked with 5% non-fat milk overnight and probed with primary antibodies against target proteins at 4 °C overnight, followed by incubation with a secondary Goat Anti-Rabbit IgG and Anti-Mouse IgG (Cell Signaling, Technology, Inc., Danvers, MA, USA) (1:20,000 dilution; Bio-Rad, Hercules, CA, USA) at 37 °C for 60 min. The following antibodies were used: anti-Bcl-xL, anti-Bcl-w, anti-Bfl-1, anti-Bcl-2, and anti-Mcl1 (diluted 1:1000; Cell Signaling Technology, Inc., Danvers, MA, USA). Target bands were visualized by enhanced chemiluminescence (ECL) solution (Bio-Rad, Hercules, CA, USA) and analyzed by Gel-Pro-Analyzer software (Media. Cybernetics, Inc., Bethesda, MD, USA). Anti-human β-tubulin and GAPDH (1:1000 dilution and 1:1000 dilution; Cell Signaling Technology, Inc., Danvers, MA, USA) was used as an internal control for the samples. 

### 4.14. Statistical Analysis

The results are presented as mean ± standard deviation (M ± SD). The experiments were carried out in at least three repetitions (*n* ≥ 3). The statistical significance of the difference was determined by a one-way ANOVA, followed by multiple Holm-Sidak comparisons, *p* < 0.05.

## Figures and Tables

**Figure 1 ijms-23-07881-f001:**
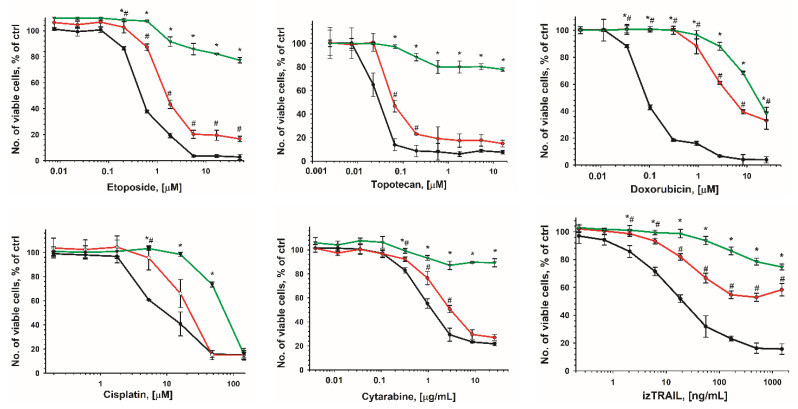
Cytotoxic effect of izTRAIL and different drugs added to THP-1 cell cultures in 1 day (black line); 3 days (red line); 5 days (green line) after cell seeding. Data are presented as mean ± standard deviation (*n* ≥ 3). *, # *p* < 0.05, compared to THP-1 cell cultures in 1 day.

**Figure 2 ijms-23-07881-f002:**
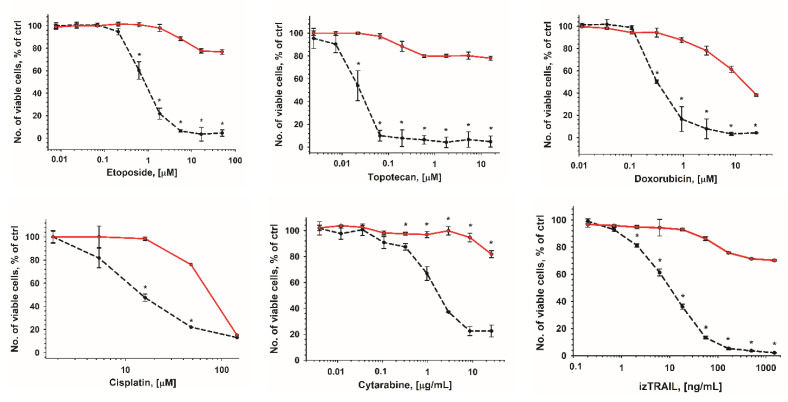
Loss of increased resistance of THP-1HDC cells (red line) after seeding them in low-density culture (5000 cells/well) and 1 day of cultivation (black dotted line). Data are presented as mean ± standard deviation (*n* ≥ 3). * *p* < 0.05, compared to THP-1HDC cells.

**Figure 3 ijms-23-07881-f003:**
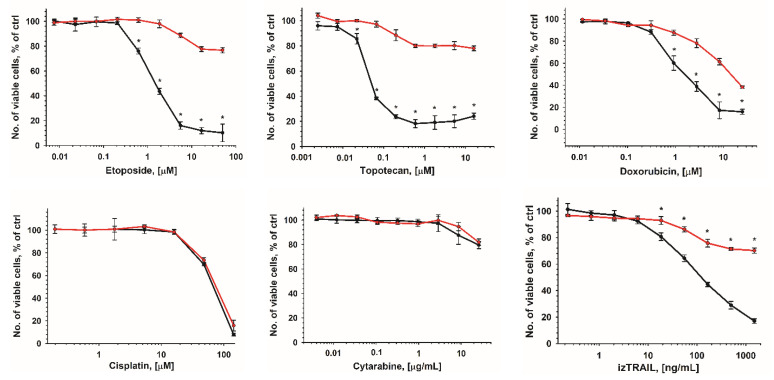
Comparison of the cytotoxicity of drugs in THP-1 cell cultures that reached a density of 10^6^ cells/mL at 5 days after seeding of 5 × 10^4^ cells/mL (THP-1HDC, red line) and at 1 day after seeding of 5 × 10^5^ cells/mL (THP-1HDC_1day_, black line). * *p* < 0.05, compared to THP-1HDC cells.

**Figure 4 ijms-23-07881-f004:**
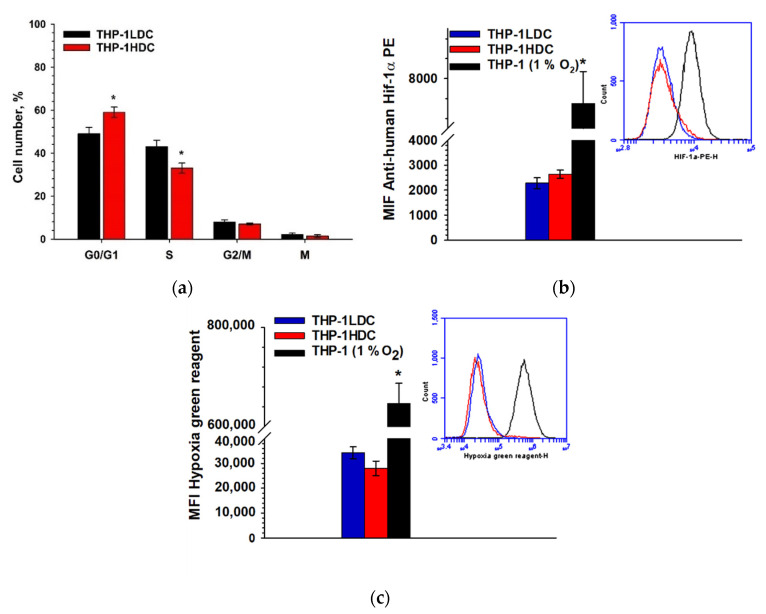
Analysis of cell distribution by cell cycle phases (**a**) and hypoxia in THP-1HDC cells. Mean intensity of fluorescence (MIF) of anti-human HIF-1α PE (**b**) and hypoxia green reagent (**c**) in THP-1LDC, THP-1HDC, and THP-1LDC1%O_2_ (THP-1 (1% O_2_)) cells. Data are given as mean ± standard deviation (*n* ≥ 3). * *p* < 0.05, compared to THP-1LDC cells.

**Figure 5 ijms-23-07881-f005:**
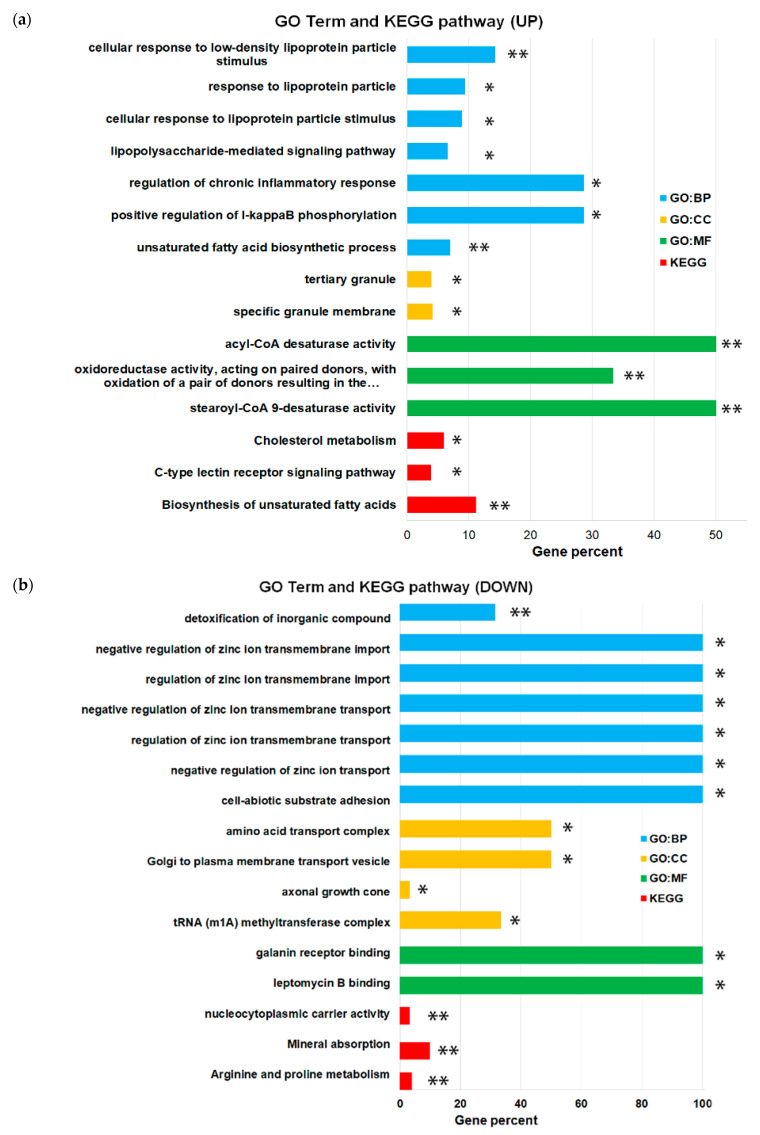
GO-term analysis of the BP, CC, and MF categories enrichment and the KEGG pathway analysis with the 121 differentially expressed genes in THP-1HDC cells, compared to THP-1LDC cells for up-regulated genes (**a**) and down-regulated genes (**b**). Statistical significance was determined using the Bonferroni step-down test and a two-tailed hypergeometric enrichment/depletion test. * *p* < 0.05; ** *p* < 0.001.

**Figure 6 ijms-23-07881-f006:**
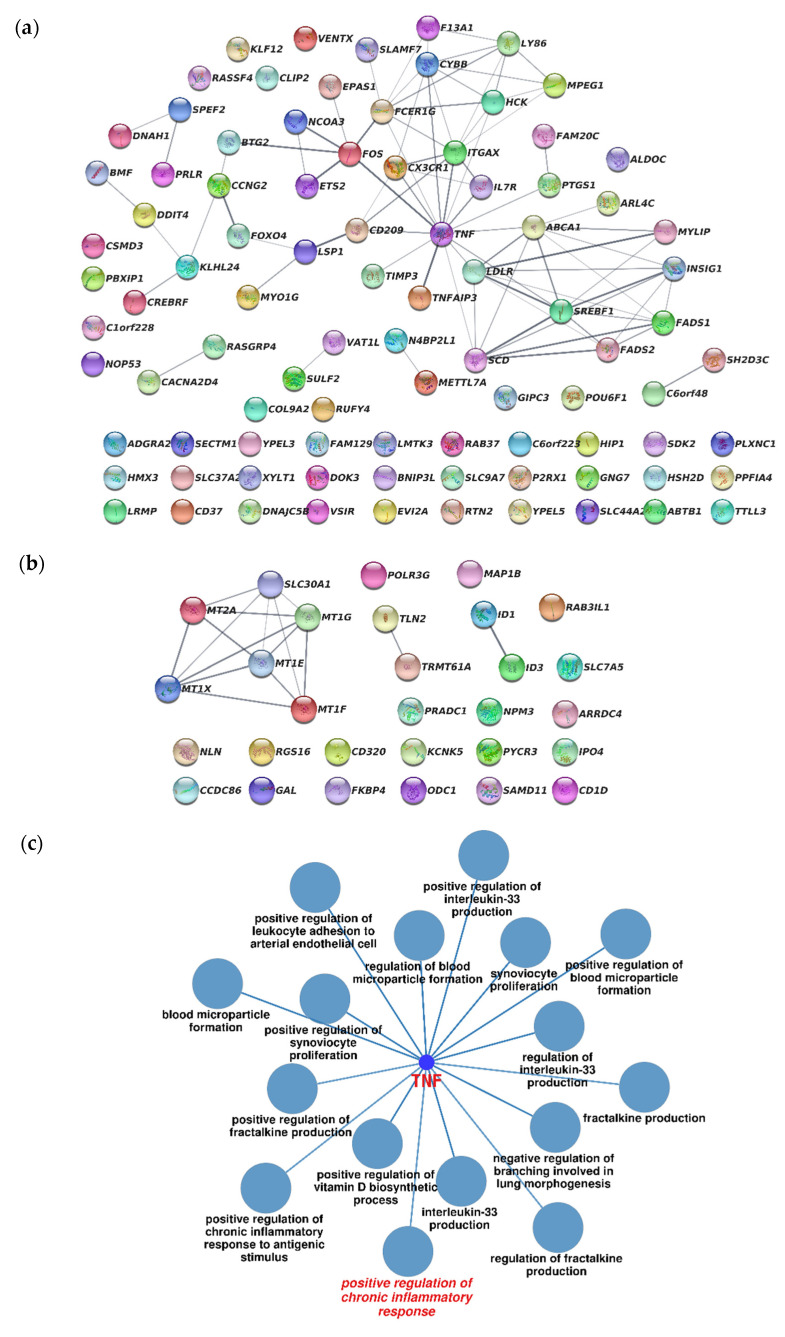
Analysis of the PPI network of up-regulated (**a**) and down-regulated (**b**) genes and functional analysis of BP annotations of GO, which are connected with TNFa, the hub node of PPI network in THP-1HDC cells (**c**). PPI, protein–protein interaction.

**Figure 7 ijms-23-07881-f007:**
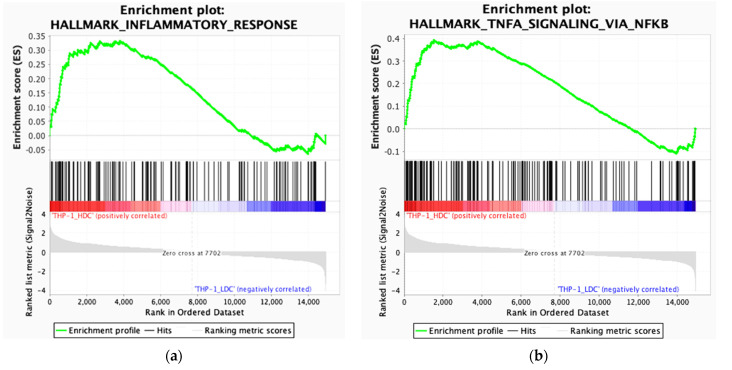
GSEA analysis by Hallmark gene sets for inflammatory response (ES = 0.333, NES = 1.589) (**a**) and TNF via NF-kB (ES = 0.393, NES = 1.930) (**b**), *p*-value ≤ 0.05. NES—normalized enrichment score.

**Figure 8 ijms-23-07881-f008:**
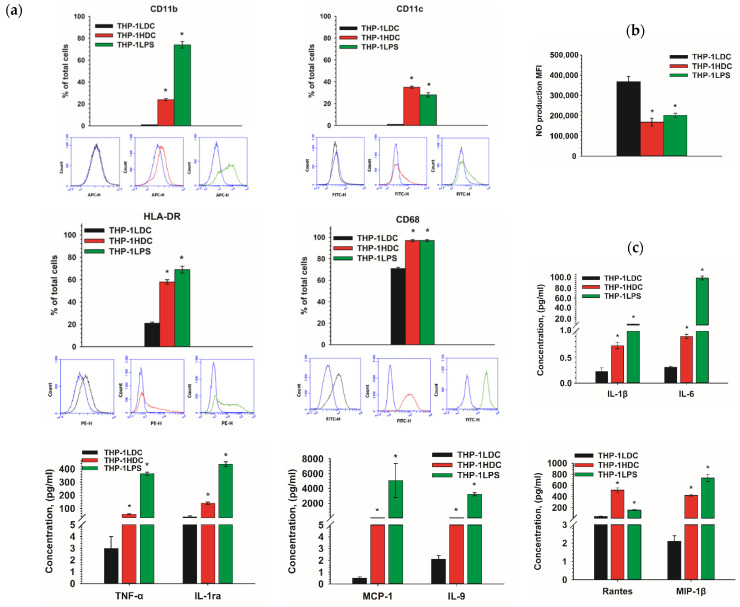
Immunophenotype analysis (**a**) and analysis of intracellular nitrogen production (**b**) in THP-1HDC, THP-1LDC, and THP-1LPS cells. The flow cytometry histograms are presented for the cells stained with IG Isotypes (blue curves), and for THP-1LDC (black), THP-1HDC (red), and THP-1LPS (green) cells stained with corresponding antibodies. Analysis of the cytokine secretion by THP-1LDC, THP-1HDC, and THP-1LPS cells (**c**). (Y-axis—concentration of cytokines in the culture medium, normalized to 1 × 10^5^ cells/mL). The data are given as mean ± standard deviation (*n* ≥ 3). * *p* < 0.05, compared to THP-1LDC cells.

**Figure 9 ijms-23-07881-f009:**
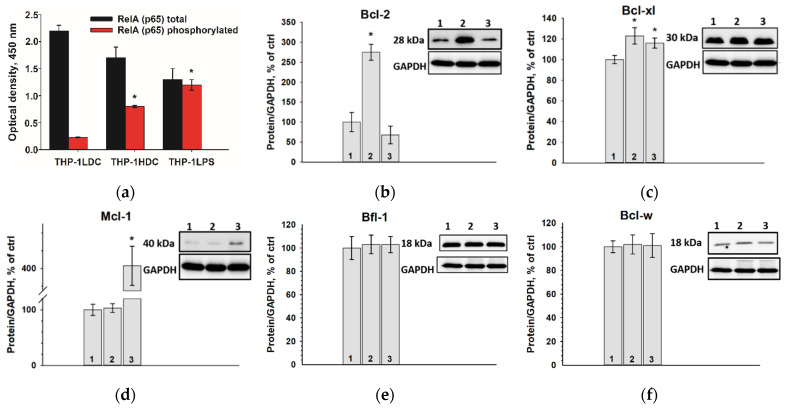
Increase in the content of phosphorylated RelA (p65) in THP-1HDC and THP-1LPS cells, compared to THP-1LDH cells (**a**). Content of Bcl-2 (**b**); Bcl-xL (**c**); Mcl-1 (**d**); Bfl-1 (**e**); and Bcl-w (**f**) proteins in THP-1HDC cells (column 2) and THP-1LPS (column 3), compared to THP-1LDC cells (column 1). GAPDH was used as a loading control for protein normalization. The protein content in the cell lysate without any additives served as a control (100%). Data are given as mean ± standard deviation (*n* = 3). * *p* < 0.05 compared to THP-1LDC cells.

**Figure 10 ijms-23-07881-f010:**
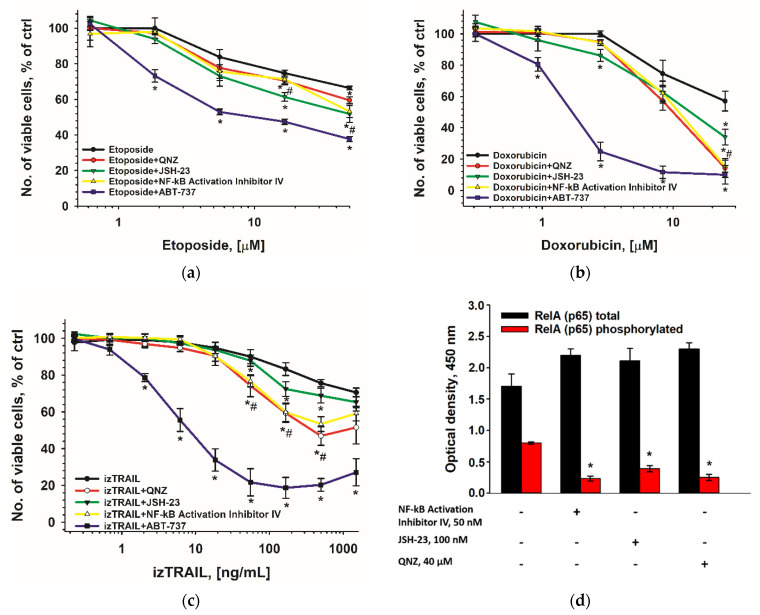
Effect of the inhibitors of NF-kB (JSH-23 (100 nM); QNZ (40 μM); and NF-kB Activation Inhibitor IV (50 nM)), and of the anti-apoptotic proteins of the Bcl-2 family (ABT-737, 3 μM) on drug resistance of THP-1HDC cells (**a**–**c**). *^,^# *p* < 0.05 compared to control (cells treated with DNA-damaging agent only or izTRAIL). Decrease in the level of phosphorylated RelA (p65) in THP-1HDC cells after 4-h incubation them with the NF-kB inhibitors (**d**). Data are presented as the mean ± standard deviation. *n* = 3. * *p* < 0.05 compared to control (untreated cells).

**Figure 11 ijms-23-07881-f011:**
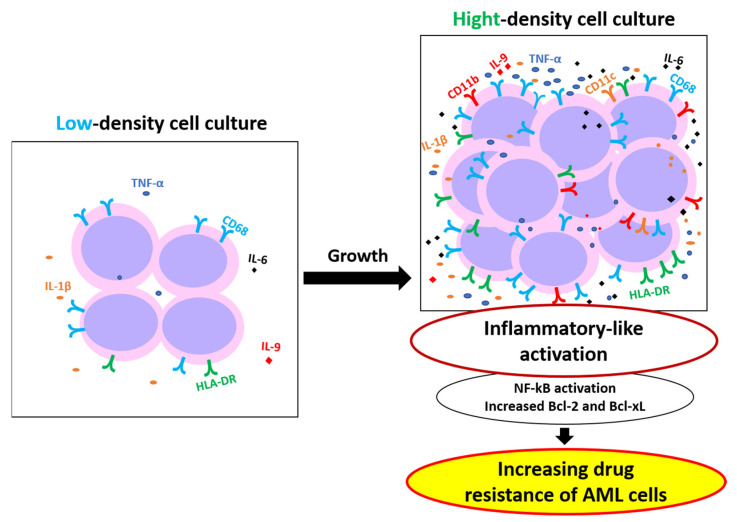
Proposed mechanism for increased resistance of AML cells in high-density cell cultures.

## Data Availability

Not applicable.

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
