# Peer review of "The Increase in the Drug Resistance of Acute Myeloid Leukemia THP-1 Cells in High-Density Cell Culture Is Associated with Inflammatory-like Activation and Anti-Apoptotic Bcl-2 Proteins"

_ijms, 2022, doi:10.3390/ijms23147881_

Round 1
Reviewer 1 Report
The article by Fadeev et. al nicely illustrates the mechanism behind the increase in drug resistance of acute myeloid leukemia (AML) cells in high-density cell culture. The thorough studies performed by the authors indicate the close connection of this phenomenon with inflammatory cytokines and anti-apoptotic proteins. Furthermore, an improvement in suppressing drug resistance in high-density AML cells was observed when inhibitors of NF-kB and Bcl-2 were used. In concert, rationally designed studies by the authors described in this work hold the potential of contributing significantly toward the understanding of AML phenotype and treatment strategies. For these reasons, this work is suitable for publication in IJMS. It meets the impact criteria of this journal because the work will be of significant interest to a broad readership in the chemical biology and medicinal chemistry community. However, several minor issues should be addressed beforehand, as listed below.
1) The reviewer couldn’t find Figure 9 in the draft. Assuming it’s an editing error, this faux pas should be rectified in the original manuscript version.
2) The reviewer is wondering if the concentration used for DNA-damaging drugs and izTRAIL was kept the same for both LDC and HDC cells. This could be important as the average number of cells/drug solutions will significantly vary otherwise.
3) References should be added in some places (e.g., at the end of the line 271 statement). The same goes with the absence of literature citing the connection between AML and inflammatory cytokines or anti-apoptotic proteins in the draft. The introduction section requires a detailed discussion of these factors contributing to current state-of-the-art AML diagnosis.
Author Response
Dear Reviewer, thank you very much for your comments on our manuscript!
Response 1: The reviewer couldn’t find Figure 9 in the draft. Assuming it’s an editing error, this faux pas should be rectified in the original manuscript version
Point 1: Figure 9 is added in the draft.
Response 2: The reviewer is wondering if the concentration used for DNA-damaging drugs and izTRAIL was kept the same for both LDC and HDC cells. This could be important as the average number of cells/drug solutions will significantly vary otherwise.
Point 2: The concentration of DNA-damaging drugs and izTRAIL was kept the same for both LDC and HDC cells. This point of view is supported, in particular, by the fact that the addition of ABT-737 at a non-toxic dose restored the sensitivity of HDC cells to drugs and izTRAIL (Figure 15). Another result supporting the view is that THP-1HDC1day cells appeared more high sensitivity to the drugs compared to THP-1HDC cells, despite the equal cell concentration in THP-1HDC and THP-1HDC1day cultures (Figure 4)
Response 3: References should be added in some places (e.g., at the end of the line 271 statement). The same goes with the absence of literature citing the connection between AML and inflammatory cytokines or anti-apoptotic proteins in the draft. The introduction section requires a detailed discussion of these factors contributing to current state-of-the-art AML diagnosis.
Point 3: The references are added. Recommended information is included in the introduction.
Reviewer 2 Report
The authors demonstrated that The increase in drug resistance of acute myeloid leukemia THP-1 cells in high-density cell culture was associated with in- 3 flammatory-like activation and anti-apoptotic Bcl-2 proteins. I highly appreciate the authors' effort to provide sufficient experimental evidences with careful experimental design. I just point out several issues on RNA-seq analysis.
1. The authors should disclose the raw datasets of RNA-seq.
2. The experimental schemes of RNA-seq (e.g., what kind of samples, how many samples) are unclear. The authors should explain the details.
3. The authors should show the statistical significance (not only Gene percent) of GO and KEGG pathway enrichment analysis in Figure 7.
4. Also, the authors should perform clustering and/or PCA analysis on the RNA-seq datasets.
Author Response
Dear Reviewer, thank you very much for your comments on our manuscript!
Response 1: The authors should disclose the raw datasets of RNA-seq.
Point 1: The raw datasets of RNA-seq are disclosed in https://docs.google.com/spreadsheets/d/1sJGfDrtAwSeyMFFK1s_BZzYc5IbiU6tD/edit?usp=sharing&ouid=111315584677286643095&rtpof=true&sd=true
Response 2: The experimental schemes of RNA-seq (e.g., what kind of samples, how many samples) are unclear. The authors should explain the details.
Point 2: The explanations are added in the section materials and methods.
Response 3: The authors should show the statistical significance (not only Gene percent) of GO and KEGG pathway enrichment analysis in Figure 7.
Point 3: The statistical significance is shown.
Response 4: Also, the authors should perform clustering and/or PCA analysis on the RNA-seq datasets.
Point 4: The results of PCA analysis on the RNA-seq datasets are included in the manuscript.
Round 2
Reviewer 2 Report
The authors revised the manuscript sufficiently.
Author Response
Dear Reviever, thank you very much for your feedback on our manuscript!